# High-Risk Lineages of Hybrid Plasmids Carrying Virulence and Carbapenemase Genes

**DOI:** 10.3390/antibiotics13121224

**Published:** 2024-12-17

**Authors:** Valeria V. Shapovalova, Polina S. Chulkova, Vladimir A. Ageevets, Varvara Nurmukanova, Irina V. Verentsova, Asya A. Girina, Irina N. Protasova, Victoria S. Bezbido, Victor I. Sergevnin, Irina V. Feldblum, Larisa G. Kudryavtseva, Sergey N. Sharafan, Vladislav V. Semerikov, Marina L. Babushkina, Inna R. Valiullina, Nikita S. Chumarev, Guzel S. Isaeva, Natalya A. Belyanina, Irina U. Shirokova, Tatiana M. Mrugova, Elena I. Belkova, Svetlana D. Artemuk, Aleksandra A. Meltser, Marina V. Smirnova, Tatyana N. Akkonen, Nataliya A. Golovshchikova, Oleg V. Goloshchapov, Alexey B. Chukhlovin, Lubov N. Popenko, Elena Y. Zenevich, Aleksandr A. Vlasov, Galina V. Mitroshina, Marina S. Bordacheva, Irina V. Ageevets, Ofeliia S. Sulian, Alisa A. Avdeeva, Vladimir V. Gostev, Irina A. Tsvetkova, Maria A. Yakunina, Ekaterina U. Vasileva, Alina D. Matsvay, Dmitry I. Danilov, Yulia A. Savochkina, German A. Shipulin, Sergey V. Sidorenko

**Affiliations:** 1Centre for Strategic Planning, of the Federal Medical and Biological Agency, Moscow 119121, Russia; vnurmukanova@cspfmba.ru (V.N.); amatsvay@cspfmba.ru (A.D.M.); ddanilov@cspmz.ru (D.I.D.); ysavochkina@cspmz.ru (Y.A.S.); shipulin@cspmz.ru (G.A.S.); 2Pediatric Research and Clinical Center for Infectious Diseases, Saint Petersburg 197022, Russia; ageevets.va@niidi.ru (V.A.A.); ageevets.iv@niidi.ru (I.V.A.); sulyan.os@niidi.ru (O.S.S.); avdeeva.aa@niidi.ru (A.A.A.); vgostev@niidi.ru (V.V.G.); itsvetkova@niidi.ru (I.A.T.); sidorserg@niidi.ru (S.V.S.); 3Khanty-Mansyisk Regional Clinical Hospital, Khanty-Mansiysk 622018, Russia; verentsovaiv@okbhmao.ru; 4Khanty-Mansiysk State Medical Academy, Khanty-Mansiysk 628011, Russia; aa.girina@hmgma.ru; 5Department of Microbiology Named After Associate Professor B.M. Zelmanovich, Krasnoyarsk State Medical University Named After Professor V. F. Voyno-Yasenetsky, Krasnoyarsk 660022, Russia; i.protasova@cspfmba.ru; 6Krasnoyarsk Interdistrict Clinical Emergency Hospital Named After N.S. Karpovich, Krasnoyarsk 660062, Russia; v.bezbido@cspfmba.ru; 7Department of Epidemiology and Hygiene, Perm State Medical University Named After Academician E.A. Wagner, Perm 614000, Russia; v.sergevnin@cspfmba.ru (V.I.S.); i.feldblum@cspfmba.ru (I.V.F.); 8Federal Center for Cardiovascular Surgery Named After S.G. Sukhanov of the Ministry of Health of the Russian Federation, Perm 614013, Russia; klg@permheart.ru (L.G.K.); shsn@permheart.ru (S.N.S.); 9State Budgetary Healthcare Institution of Perm Krai «Perm Regional Clinical Infectious Diseases Hospital», Perm 614014, Russia; v.semyarkov@cspfmba.ru (V.V.S.); babushkina@kkib.ru (M.L.B.); 10Republican Clinical Hospital of the Ministry of Health of the Republic of Tatarstan, Kazan 420064, Russia; inna.valiulleena@tatar.ru; 11Department of Microbiology Named After Academician V.M. Aristovsky, Kazan State Medical University of the Ministry of Health of the Russian Federation, Kazan 420012, Russia; nikita.chumarev@kazangmu.ru (N.S.C.); guzelle.isaeva@kazangmu.ru (G.S.I.); 12Kazan Scientific Research Institute of Epidemiology and Microbiology, Kazan 420015, Russia; 13Department of Epidemiology, Microbiology and Evidence-Based Medicine, Federal State Budgetary Educational Institution of Higher Education «Privolzhsky Research Medical University» of the Ministry of Health of the Russian Federation (FSBEI HE PRMU MOH Russia), Nizhny Novgorod 603005, Russia; belyanina_n@pimunn.net (N.A.B.); shirokova_i@pimunn.net (I.U.S.); 14State Budgetary Institution of Healthcare of the City of Moscow «Moscow Science and Practical Centre for the Laboratory Research of the Department of Healthcare of the City of Moscow», Moscow 127015, Russia; mrugovatm@dcli.ru; 15Mariinsky Hospital, Saint Petersburg 191014, Russia; bak@mariin.ru (E.I.B.); s.artemuk@cspfmba.ru (S.D.A.); a.meltzer@mariin.ru (A.A.M.); m.smirnova@cspfmba.ru (M.V.S.); 16City Polyclinic No. 74, Saint Petersburg 197762, Russia; t.akkonen@cspfmba.ru (T.N.A.); n.golovshchikova@cspfmba.ru (N.A.G.); 17R. M. Gorbacheva Memorial Institute of Oncology, Hematology and Transplantation, Pavlov University, Saint Petersburg 197022, Russia; o.goloshchapov@cspfmba.ru (O.V.G.); a.chukhlovin@cspfmba.ru (A.B.C.); 18I. I. Dzhanelidze Research Institute of Emergency Medical Service, Saint Petersburg 192242, Russia; baklab@emergency.spb.ru; 19Alexander Hospital, Saint Petersburg 190068, Russia; e.zenevich@cspfmba.ru; 20Psychiatric Hospital No. 1 Named After P. P. Kaschenko, Saint Petersburg 195009, Russia; a.a.vlasov@kaschenko-spb.ru; 21Napalkov State Budgetary Healthcare Institution “Saint-Petersburg Clinical Scientific and Practical Center for Specialised Types of Medical Care (Oncological)”, Saint Petersburg 197101, Russia; nponkcentr@zdrav.spb.ru (G.V.M.); m.s.bordacheva@oncocentre.ru (M.S.B.); 22Department of Medical Microbiology, North-Western State Medical University Named After I.I. Mechnikov, Saint Petersburg 191015, Russia; 23Murmansk Regional Clinical-Hospital Named After P.A. Bayandin, Murmasnk 183047, Russia; yakunina@mokb51.ru (M.A.Y.); vasilevaeu@mokb51.ru (E.U.V.)

**Keywords:** *Enterobacterales*, carbapenem resistance, virulence, plasmids

## Abstract

**Background/Objectives:** Carbapenem-resistant *Enterobacterales* (CRE) are a global health threat due to their high morbidity and mortality rates and limited treatment options. This study examines the plasmid-mediated transmission of virulence and antibiotic resistance determinants in carbapenem-resistant *Klebsiella pneumoniae* (*Kpn*) and *Escherichia coli* (*E. coli*) isolated from Russian hospitals. **Methods**: We performed short- and long-read whole-genome sequencing of 53 clinical isolates (48 *Kpn* and 5 *E. coli*) attributed to 15 genetic lineages and collected from 21 hospitals across nine Russian cities between 2016 and 2022. **Results**: The plasmid analysis identified 18 clusters that showed high concordance with replicon typing, with all clusters having a major replicon type. The majority of plasmids in the IncHI1B(pNDM-MAR)/IncFIB(pNDM-Mar)-like cluster (79.16%) carried both antibiotic resistance genes (e.g., *bla*_NDM-1_ and *bla*_OXA-48_) and virulence factors (VFs) such as siderophore genes. We hypothesized that hybrid plasmids could play a critical role in the dissemination of antibiotic resistance genes and VFs. Comparative analyses with global plasmid databases revealed high-risk lineages of hybrid plasmids that are predominantly spread throughout Russia at present. **Conclusions:** Our findings underscore the importance of monitoring plasmid backbones for clinical management, surveillance, and infection control activities.

## 1. Introduction

Drug-resistant bacteria pose a threat to individual and public health worldwide. Carbapenem-resistant *Enterobacterales* (CRE) have been listed by the World Health Organization (WHO) as one of the top three classes of drug-resistant bacteria in the world that urgently require the development of new antibiotics [1]. Infections caused by CRE tend to be more severe, with significantly higher morbidity and mortality when compared with those caused by susceptible bacteria [2]. Almost 126,000 (125,950) global deaths were attributed to antimicrobial resistance of CRE in 2019 [3]. Different aspects of the CRE epidemiology have been studied worldwide, with a focus on the most common carbapenem-resistant *Klebsiella pneumoniae* (*Kpn*) and *Escherichia coli* (*E. coli*), which, together, account for more than 90% of CRE strains and are widely spread worldwide through multiple routes [4]. Carbapenem resistance can be conferred by vertically inherited point mutations or via the acquisition of horizontally transmitted genes, often located in mobile genetic elements ((MGEs) e.g., plasmids, transposons, and integrons). Plasmids play an important role in the dissemination of genes encoding carbapenem-hydrolyzing enzymes (carbapenemases) among different bacterial species [5]. Additional genes conferring resistance to other antibiotics, such as aminoglycosides and fluoroquinolones, can also be carried by these elements, further limiting the treatment options.

Virulence factors (VFs), which can also be transferred by plasmids, additionally contribute to pathogenicity by enhancing the virulence potential of bacteria. There is evidence that hypermucoidy and enhanced iron acquisition via siderophore production play roles in the hypervirulent pathotype of *Kpn* [6]. The gene clusters encoding these features (*iuc*, *iro*, *rmpADC*, and its ortholog *rmpA2D2C2*) are usually carried on large so-called virulence plasmids. Plasmids pK2044 and pLVPK from *Kpn* strains are well-studied, non-conjugative, conservative virulence plasmids that have long been considered specific to community-acquired infections in the Asia–Pacific region. The transmission of the virulence plasmid to other *Enterobacterales* has been reported. The pLVPK-like plasmid was revealed in an *E. coli* foodborne strain (isolated from chicken meat) in China. The authors also indicated that aerobactin and salmochelin gene loci were widespread in other *E. coli* strains in China (22–48%) [7].

For a long time, multidrug resistance and hypervirulence were deemed as two non-overlapping pathotypes of *Kpn*. Strains that cause ‘classical’ infections in hospitals—often with high rates of multidrug resistance—were named classical *K. pneumoniae* (cKp), and strains that cause drug-susceptible ‘hypervirulent’ infections in community settings were named hypervirulent (hvKp). However, in recent years, there has been an increasing number of reports on the convergence of these pathotypes and their genetic determinants, resulting in devastating clinical outcomes [8]. The formation of a convergent pathotype—that is, combining MDR and hypervirulence determinants—is the current evolutionary stage of *Kpn,* which emerged in 2018. Since then, their incidence among various genetic lineages has been increasing globally, which poses a serious threat to the healthcare system. The formation of convergent *Kpn* is associated with the transition of hypervirulence genes from non-conjugative plasmids into new plasmid backbones and, then, into new *Kpn* genetic lineages (including European ones). Thus, the spread of resistance genes among Asian genetic lines (CG23) and the main virulence genes among genetic lines of MDR *Kpn* (ST395, ST147, etc.) was revealed. The mobilization of plasmids with virulence markers led to enhanced horizontal transfer of genes associated with hypervirulence and, as a result, to the formation of new genetic environments. There have been many reports on the acquisition of a single vector carrying a combination of determinants of two *Kpn* pathotypes, including resistance genes to various classes of antibiotics. Nevertheless, the combination of carbapenem resistance (CR) genes with virulence determinants is of the greatest concern due to the use of the latter in the treatment of severe infections. The number of large plasmids carrying hypervirulence genes, which often include multiple replicons, has sharply increased. All of them share sequence fragments of the canonical pLVPK plasmid. To date, hybrid plasmids carrying virulence and carbapenemase genes have spread throughout various genetic lines. Their relevance is underlined by a very recent report (published in July 2024 by the WHO) concerning the global risk of increased identification of isolates of hvKp sequence type (ST) 23 carrying carbapenem antibiotics resistance genes (CR-hvKp) [9]. Complete and accurate genome assemblies, which can be produced via hybrid genome assembly of short- and long-read sequencing data, are crucial for the study of plasmid epidemiology. While the majority of studies analyzing human-associated isolates have focused only on resistance determinants and their vehicles, other genetic elements, such as VFs located on the plasmids, should also be monitored. In this study, we examined the plasmidome of CRE clinical isolates of different STs via genomic analysis, focusing on those sequences that might be related to the transfer of VFs.

## 2. Results

A total of 53 genomes were included in this study, which were sequenced using the Nanopore platform: 48 *Kpn* and 5 *E. coli* (Appendix A). According to an analysis using the MOB-suite reconstruct option, the median number of plasmids in *Kpn* genomes was 3 (min. 1, max. 8), and that in *E. coli* genomes was 5 (min. 5, max. 6). Circular contigs (n = 193) that were classified as originating from plasmids were clustered using the mge-cluster tool to describe the plasmidome in all isolates (Table 1). It was found that 73.3% of plasmids coded one replicon gene, whereas 26.67% coded more than one replicon. Clustering into 18 clusters showed high concordance with replicon typing, with all clusters having major replicon type (Appendix A), and 7 out of 18 clusters consisted of sequences with a predominant replicon type of family Col [ColRNAI, Col(pHAD28), Col440II, Col(MG828)]. The largest cluster (n = 48, cluster 2) consisted of plasmids with IncHI1B(pNDM-MAR) and IncFIB(pNDM-Mar) replicons.

There were eight clusters formed by plasmids isolated only from *Kpn*, including IncFIBK-like (13 cluster), Col-like (5 and 6 clusters), IncL-like (7 cluster), and others; whereas only one cluster (14) consisted of plasmids from *E. coli*, which was predominantly formed by plasmids with IncI(Gamma), IncFIA, and IncI1-I(Alpha) replicons. Nearly half of all plasmids (n = 95) were predicted to be conjugative, and each cluster tended to be formed by plasmids with the same type of mobility.

Conjugative plasmids had the largest median length (160,194 bp), compared with those of mobilizable (9294 bp) and non-mobilizable (7462 bp) plasmids (Appendix A). Large (>250 kb) plasmids (n = 40) were identified only in the cluster with IncHI1B(pNDM-MAR)/IncFIB(pNDM-Mar)-like plasmids, and almost all of them were conjugative.

ARGs were found in 11 clusters, of which 5 clusters had plasmids with carbapenemase genes (Figure 1, Appendix A): *bla*_NDM-1_ (n = 27), *bla*_OXA-48_ (n = 21), *bla*_KPC-3_ (n = 4), *bla*_OXA-244_ (n = 3), *bla*_NDM-5_ (n = 2), and *bla*_NDM_ (n = 1, pVKpST874_2270).

While OXA-48-like carbapenemases were predominantly carried by IncL-like plasmids, the majority of NDM carbapenemases were identified on IncHI1B(pNDM-MAR)/IncFIB(pNDM-Mar)-like plasmids. In contrast to the wide distribution of ARGs among clusters, VFs were found only in two clusters (Figure 1). The aerobactin lineage iuc5 was identified only on one plasmid from *E. coli*, with IncFIB(AP001918), IncFIC(FII), and IncFIA replicons. The aerobactin lineage iuc1, *rmpA/A2*, *peg-344*, and *iroBCDN* genes were identified on IncHI1B(pNDM-MAR)/IncFIB(pNDM-Mar)-like plasmids (44/48, 91.67%). The majority (38/48, 79.16%) of IncHI1B(pNDM-MAR)/IncFIB(pNDM-Mar)-like plasmids were hybrid, as they carried both ARGs and VFs.

Five highly similar plasmids from the IncHI1B(pNDM-MAR)/IncFIB(pNDM-Mar)-like cluster were chosen for multiple sequence alignment (Figure 2A).

Notably, one of them (pVKpST512_3061) carried neither ARGs nor aerobactin. Despite the difference in coding of some genes, there were many highly similar regions. The tellurite resistance system—which plays a role in protecting bacteria from host defenses—was identified in all cases. A similar ARG cassette array of composite transposone and class 1 integron, which consisted of genes coding for chloramphenicol O-acetyltransferase (catA1), aminoglycoside-2″-O-nucleotidyltransferase (ant(2″)-Ia), aminoglycoside (3″) (9) adenylyltransferase (aadA1), disinfectant resistance protein (qacEdelta1), and sulfonamide-resistant gene (sul1), was present on both ARG-carrying plasmids: TnAs3-ant(2″)-Ia-aadA1-qacEdelta1-sul1-TnAs3. TnAs3 has been previously associated with cat genes that are located upstream of the array. The virulence fragment, which contained aerobactin cluster genes, was flanked by IS630 (upstream) and ISEc16 and ISKqu3 (downstream). This fragment, located between common sequences containing IS1A and ISYps3, was absent in pVKpST512_3061, which implies the genetic transposition of this locus (Figure 2B). Notably, clusters of genes responsible for defense systems were identified on this plasmid: CAS_Class1-Subtype-IV-A, Mok_Hok_Sok toxin–antitoxin system, and RM_Type_II.

IncHI1B(pNDM-MAR)/IncFIB(pNDM-Mar)-like plasmids (cluster 2) were compared with highly similar complete plasmids downloaded from the NCBI database (Appendix A). According to the metadata provided in genbank files, the downloaded plasmids were collected in 26 different countries. Four clusters of plasmids [A (n = 344), B (n = 67), C (n = 26), D (n = 5)] were obtained and visualized using networks (Figure 3A).

Almost all plasmids (87.78%, 388/442) were from humans, but there were five plasmids from the environment (the telephone of a nurse station, a ventilator, bed sheets, and a monitor panel) and one plasmid from a cow. The majority of plasmids (93.67%, 414/442) were isolated from the *Klebsiella* complex ((*Kpn*) n = 412; *K. quasipneumoniae*, n = 2), while there were 27 plasmids from *E. coli* and 1 plasmid from *Enterobacter hormaechei*.

The largest cluster A (the “Chinese” cluster) consisted of plasmids that originated from 24 different countries, but a large proportion (238, 69.19%) of plasmids were from China (Figure 3B). Only five plasmids from this cluster were from Russia (four of which were sequenced in this study). Plasmids from this cluster were isolated from different lineages of *Kpn* and *E. coli*, but some communities (Appendix A) were predominantly formed by plasmids isolated from the same sequence type, such as ST11 (n = 121), ST23 (n = 59), ST86 (n = 12), ST65 (n = 13), ST15 (n =7), and ST10 (n = 11) from *E. coli* (Appendix A). All plasmids of *E. coli* ST10 were isolated between 2022 and 2023 in China. In contrast to cluster A, plasmids from cluster B (the “European” cluster) were not widely spread (nine countries, six of which were in Europe). Nearly half of these plasmids were from Russia (n = 32, 47.76%), while almost one quarter (24.6%) were from the U.K. Major sequence types of clusters B and C were ST147 and ST395 (Appendix A). Small clusters C and D were almost all formed by plasmids from Russia. There was one plasmid from China (NZ_CP061962, year 2015) and one from Ukraine (NZ_CP132661, year 2015) in cluster C. All plasmids from cluster D were isolated from ST377.

The majority of analyzed plasmids from this dataset (n = 380, 85.97%) were multiple replicons. Almost 90% of plasmids (n = 396, 89.59%) coded the IncHI1B(pNDM-MAR) replicon, which was identified as a single replicon. Interestingly, the exact sequence of this replicon had some association with detected communities (Figure 3C). One sequence, labeled “a” in Figure 3C, significantly differed from the reference sequence for the IncHI1B(pNDM-MAR) replicon; in particular, the leading 97 nt was unaligned using BLAST [10]. The majority of plasmids in cluster A, in addition to the IncHI1B(pNDM-MAR) replicon, coded another replicon gene, repB_KLEB_VIR, and sometimes included an additional third or even fourth replicon [IncI1-I(Alpha), ColRNAI, IncM2, IncFIB(pKPHS1), IncR, IncFII(pHN7A8), IncFIA(HI1), IncU, IncN, IncX3] (Figure 3D). Plasmids from clusters B and C coded the IncHI1B(pNDM-MAR) replicon, often with an additional IncFIB(pNDM-Mar) replicon. All plasmids from a small cluster D coded the replicon IncFIB(K)(pCAV1099-114).

Most plasmids from cluster A were non-mobilizable and did not code ARGs (Figure 3E,F), whereas plasmids from clusters B and D were conjugative and coded carbapenemases (*bla*_NDM_ and *bla*_OXA-48_). All the cluster A plasmids carried aerobactin, except for six non-mobilizable (also without ARGs) plasmids that did not, one of which (NZ_LR890425)—which was isolated from a human (blood) in Australia in 2014—had high similarity to our plasmid pVKpST86_86 (also non-mobilizable, without ARGs; Figure 4).

In addition, these six plasmids coded *rmpA* and *peg-344*. These plasmids carried the same cointegrated replicon as the canonical virulent plasmid pLVPK, which was also present in clusters carrying the repB_KLEB_VIR replicon (cluster A). In fact, this group of plasmids can be attributed to the precursors in the spread of virulence markers. Four plasmids from clusters B and C did not code for aerobactin, and none of the plasmids from cluster D coded any VFs.

## 3. Discussion

Understanding the evolution and spread of antibiotic-resistant microorganisms is crucial to countering the serious global threat posed by these organisms. Our dataset consisted of *Kpn* and *E. coli* isolates that were resistant to carbapenems—vital last-resort antibiotics against severe bacterial infections unresponsive to common drugs. Exploring genetic factors that might play a key role in enabling these strains to induce illness in humans is essential. It has recently been shown that VFs located on large virulence plasmids such as *rmpADC*, *iuc*, *iro*, and *peg-344* could be considered as markers of *Kpn* with a hypervirulent phenotype [11,12]. The development and dissemination of methods such as Oxford nanopore long-read sequencing have enabled the generation of complete plasmid sequences, which is crucial for detecting and tracking the spread of ARGs and VFs across communities.

In this study, we characterized the plasmidome of CR clinical *Kpn* and *E. coli* with a focus on complete plasmids carrying VFs. A total of 18 clusters of plasmids were identified, demonstrating remarkable genetic heterogeneity. However, plasmids carrying ARGs and/or VFs were detected in only five of those clusters. The majority of carbapenemases were identified in two clusters: IncHI1B(pNDM-MAR)/IncFIB(pNDM-Mar)-like (cluster 2) and IncL-like (cluster 7). Medium-sized (median length 65,474 bp) IncL-like plasmids have been globally reported to be associated with *bla*_OXA-48_ [13], although this gene was also detected on IncFIA(HI1) plasmids in our dataset. Large IncHI1B(pNDM-MAR)/IncFIB(pNDM-Mar)-like plasmids carried mostly the *bla*_NDM-1_ gene, but *bla*_OXA-48_ carbapenemase was also found. More importantly, these plasmids from the IncHI1B(pNDM-MAR)/IncFIB(pNDM-Mar)-like cluster (named the virulence cluster later in the text) were the main source of VFs, including possible markers (*rmpADC*, *iuc*, *iro*, and *peg-344*) of hypervirulence in *Kpn*. It is important to note that the detection of certain VFs does not guarantee a cell’s ability to cause an hvKp-specific infection. Most of these plasmids were predicted to be conjunctive and coded multiple replicons, such as IncH1B(pNDM-MAR) and IncFIB(pNDM-MAR). They probably play a key role in both processes: the spread of CR genes (e.g., *bla*_NDM-1_ and *bla*_OXA-48_) and VFs. To date, similar sequencing studies have focused on plasmids carrying carbapenemases and/or VFs throughout various genetic lines in Europe (Italy, Germany, Czech Republic, Switzerland, Denmark, and U.K.) [14], Asia (China and Japan) [15,16], and the Middle East [17], including Egypt [18]. In Russia, the emergence of CR-hvKp has been reported among different genetic lines, including ST15, ST147, ST395, ST874 [19], ST39 [20], and ST512 [21]. Therefore, it has not been fully considered how plasmids with pathogenic factors such as ARGs and VFs might be related to plasmids without such genes. To our knowledge, previous studies of plasmid epidemiology in *Enterobacterales* have primarily focused on MDR/carbapenemase/aerobactin-carrying sequences [16,17,18]. We noticed that some plasmids from the virulence cluster that did not carry ARGs and/or VFs were highly similar to those that carried these genes. This finding demonstrates that such plasmids can be either precursors of hybrid ones or the result of deletion of ARGs and VFs loci. According to a previous study [22], an adaptation enabling plasmids to coexist with successful lineages often takes place before the plasmids acquire high-risk ARGs. We hypothesize that VFs could also be considered as acquired high-risk pathogenic factors, which might have the same dissemination mechanisms. This means that, when focusing only on these genes, the opportunity for intervention might be missed; as such, new surveillance frameworks should incorporate previously unselected samples. Lipworth S. et al. [22] have suggested that studies should move from focusing on a single phenotype or gene of interest to making efforts to identify and monitor high-risk plasmid backbones. It has been suggested, in a previous study [23], to consider plasmids as groups of linked features that might jointly influence the plasmid distribution. Thus, one necessary limitation of our study was sample collection bias, as all sequenced isolates were phenotypically filtered by their resistance to carbapenems. Hence, plasmid backbones that could play an important role in the dissemination of VFs in other phenotypes of *Enterobacterales* isolates were left unexplored. Another limitation of this study was the inability to validate some characteristics of plasmids (e.g., mobility) experimentally, and prediction of mobility was performed only using computational methods. Additionally, only phenotypic resistance to carbapenems was confirmed experimentally. VFs were identified using the database of the Kleborate tool [24], which consisted of only genetic features with strong evidence of an associated phenotype in *Kpn* that confirmed clinical relevance based on published experimental data.

A dataset of plasmids that were similar to plasmids from the virulence cluster was formed, a large proportion of which were assigned to one cluster (A). These plasmids frequently shared some common features, such as originating from China, coding the repB_KLEB_VIR replicon and VFs from a group of possible markers of hypervirulence, being non-mobilizable, and being non-coding ARGs. Despite the fact that the majority of plasmids were from China, there were 23 additional countries where this plasmid backbone was detected. This might mean that this type of plasmid is the most frequently observed and most widely disseminated in the world. In contrast, we concluded that this plasmid backbone is not widely spread in Russia, as there were only five plasmids in this cluster from that country. In addition, it should be noted that there were some communities of this cluster that were formed by plasmids isolated from strains of the same ST, such as ST11 and ST15 of *Kpn* and ST10 of *E. coli*. Some of the plasmids from ST11 were hybrid (i.e., carried both VFs and carbapenemases), which is not common for this cluster. It has been recently reported that ST11 was the main ST in China, accounting for 64.2% of all CR Kpn strains, in the period 2016–2020 [25]. ST15 CR Kpn isolates with virulence plasmids have been reported in the studies [26,27]. Notably, the authors of the first study [26] found that the analyzed CR Kpn strains were not hypervirulent, despite harboring a virulence plasmid. As mentioned in the recent review, while it is likely that the virulence plasmid represents a key therapeutic target, the current data do not demonstrate that the plasmid alone is sufficient to confer hypervirulence [6]. Furthermore, genomic analysis of the plasmid replicons and carbapenemases and siderophore-encoding genes located on plasmids also revealed the potential spread of virulence plasmids isolated from ST11 and ST15 of *Kpn* from China [28].

Clusters B and C mainly consisted of conjugative hybrid plasmids carrying both carbapenemases and VFs. While cluster B was mostly comprised of plasmids from various European countries, all plasmids from cluster C originated from Russia. Two replicons were prevalent among these groups of plasmids: IncHI1B(pNDM-MAR) and IncFIB(pNDM-Mar). In this study, it was suggested that the IncFIB replicon plays a crucial role in supporting the formation and replication of virulence plasmids [29]. Plasmids from these clusters were mainly isolated from *Kpn* ST395 and ST147. The high-level endemicity of *Kpn* ST395 in hospital settings across Russia is apparent from several studies [30,31]. In one study [30], which investigated a large international collection of *Kpn* ST395 genomes, although being biased towards carbapenemase and MDR-positive isolates, at least one virulence biomarker (among aerobactin, salmochelin, regulators of mucoid phenotypes, and the metabolite transporter gene) was detected in 43.8% of the isolates. Additionally, our genomic plasmid analysis showed that each cluster (B and C) was associated with the unique sequence of the IncHI1B(pNDM-MAR) replicon. It has been suggested [28] that monitoring the core replicon signatures might assist in identifying the spread of these hybrid plasmids. Due to the relative stability of replicon sequences, they could be used as signatures of CRE isolates with virulence potential for clinical management, surveillance, and infection control activities. Our study demonstrates the spread of high-risk plasmid backbones across Russia, with a potential future dissemination of these plasmid lineages to other countries. The present study might serve as a basis for further, more detailed analyses of plasmid epidemiology at the global level. Insights into the diversity of plasmids could be provided through the future widespread application of cost-effective WGS approaches across various countries, combined with the development of novel databases and large-scale analysis tools.

## 4. Materials and Methods

### 4.1. Bacterial Strains

All isolates were from a collection of CR and/or carbapenemase-producing bacterial isolates of *Kpn* and *E. coli* referred to the laboratory. Additionally, 25 isolates of *Kpn* had a positive string test. PCR was used for the detection of the carbapenemase genes *bla*_NDM-_, *bla*_KPC-_, and *bla*_OXA-48-like_. Isolates with MIC > 0.125 mg/L and DD < 28 mm for meropenem, as well as MIC > 0.125 mg/L and DD < 25 mm for ertapenem, were classified as resistant to carbapenems. The isolates were sampled across 21 hospitals in 9 cities (St. Petersburg, Moscow, Krasnoyarsk, Perm, Khabarovsk, Nizhny Novgorod, Kazan, Murmansk, and Khanty-Mansiysk) across Russia over a 6-year period (2016–2022). In total, 1516 isolates were collected from the 21 hospitals during this period. Among these, 48 *Kpn* and 5 *E. coli* clinical isolates were available for whole-genome sequencing (WGS). Identification of bacterial species was performed using MALDI-TOF MS (Bruker Daltonics, Bremen, Germany). The collection included isolates from various sources: blood (n = 8), broncho-alveolar lavage (n = 8), urine (n = 6), sputum (n = 5), tracheal aspirate (n = 5), wound (n = 4), lung (n = 2), feces (n = 1), bile (n = 1), exudate (n = 1), and drainage (n = 1).

### 4.2. DNA Extraction, Library Preparation, and Sequencing

All isolates were cultured in LB broth at 37 °C overnight before DNA extraction. Genomic DNA was extracted using the DNeasy Blood and Tissue Kit and the QIAcube Connect Device (Qiagen, Hilden, Germany), then subjected to WGS. The isolates were sequenced using the Nanopore MinION R9 device (Oxford Nanopore Technologies, Oxford, UK, SQK-LSK109 and flow cell R9.4.1), and 31 isolates were additionally sequenced using the MiSeq/Nextseq platform (Illumina Inc., San Diego, CA, USA), as previously described [19,32]. Base-calling and demultiplexing were conducted using Guppy v5.0.16 with the ‘sup’ model.

### 4.3. Genome Assembly and Annotation

Quality assessment of raw Illumina reads, long reads, and assemblies was performed as described previously [30,32]. Kleborate v.2.3.2 [24] was used for MLST, capsule polysaccharide (K) and lipopolysaccharide (O) antigen molecular typing, and detection of virulence factors and ARGs. Additionally, AMRFinderPlus v.3.11.8 [33] was used for ARG detection. Detection of plasmid replicons was performed using the ABRicate (--min_cov 80 --min_id 90) program v.1.0.1 with the Plasmidfinder database [34]. Mob-suite v.3.1.9 [35] was used for plasmid detection and prediction of mobility.

### 4.4. Plasmid Analysis

Contigs were classified as plasmids if replicon genes were detected and/or the mob-suite tool marked this contig as a plasmid. Primary clustering was performed using mge-cluster [36] with the parameters “--perplexity 5 --min_cluster 4”, which showed the best concordance with replicon typing. All plasmids from cluster 2 were used for further analysis, and this dataset was named the “Russian dataset”. To place our “Russian dataset” in a global context, all sequences available in NCBI’s plasmid database (https://ftp.ncbi.nlm.nih.gov/refseq/release/plasmid/, accessed on 17 March 2024) were downloaded (n = 86,249). We retained all sequences (n = 55,012) that had the keywords ‘plasmid’ and ‘complete’ in the fasta header. We refer to the NCBI plasmids as the “Global dataset.” Plasmids from the “Global dataset” were consistently compared pairwise using two metrics: (i) ANI and (ii) unitig content similarity using the Jaccard coefficient. First, the pairwise ANI values were calculated using FastANI v1.34 [37], with the following options: --fragLen 200 --minFraction 0.5. Self-comparison was removed, and reciprocal pairs were merged by averaging the ANI values. Only those plasmids from the “Global dataset” that had ANI values ≥ 0.98 to plasmids from the “Russian dataset” were retained for further analysis (n = 707). All unitigs in the resulting dataset of plasmids (default k-mer 31) were identified by generating a de Bruijn graph using Bifrost [38]. Pairwise Jaccard coefficients were calculated from the unitig presence/absence matrix using the Python package sklearn v.1.5. The Jaccard coefficients were transformed into distances by subtracting from 1. Hierarchical clustering based on the obtained distances was performed using hclust and the UPGMA method, as implemented in R. The clustering results were visualized using a dendrogram and annotated using the Microreact project [39]. Network-based visualization was performed using the Python igraph package (v0.9.11). A table with pairwise Jaccard coefficients (which should be more than 0.8) was used to generate a graph. The highest Jaccard coefficient for five plasmids from the virulence cluster 2 was less than the established threshold (0.46–0.76), and so they were excluded from the network. Community detection was performed using the Leiden algorithm implemented in the Python leidenalg package (v0.10.0). All networks were visualized using the Python igraph package (v0.9.11). The comparison of the overall plasmid structures, as well as of the neighboring regions of rmp (i.e., up to 20 kbp upstream and downstream of rmp and 15 kbp for the non-coding aerobactin cluster plasmid pVKpST512_3061), was visualized with clinker v1.0 [40].

### 4.5. Data Availability

For all Illumina and MinION reads, assemblies were deposited in the NCBI database under the project accession number PRJNA1133913, and plasmid sequences from virulence cluster 2 were deposited with GenBank accession PQ126443-PQ126490 (Appendix A). The entire dataset and visualization using a dendrogram can be accessed as a Microreact project at https://microreact.org/project/3HndeKYxYKcPGDZKdkgFmu-hybridplasmidsinch1bincfib (accessed on 6 November 2024).

## 5. Conclusions

Our analysis of plasmids in CRE clinical isolates revealed a cluster of plasmids that mainly coded multiple replicons [IncHI1B(pNDM-MAR) and IncFIB(pNDM-Mar)], carbapenemase genes (*bla*_NDM-1_ and *bla*_OXA-48_), and VFs (*iuc*, *rmpA*, *rmpA2*, *peg-344*, and *iro*). The comparative analysis using a “Global dataset” containing highly identical plasmids from other countries revealed four clusters, which were named in the text according to the geographical region from which more than half of the associated plasmids were collected: namely, “Chinese”, “European”, and two “Russian” clusters. This might mean that certain backbones of these plasmids have special routes of dissemination. In particular, hybrid plasmids—which might be a key element in the formation of convergent pathotypes in *Kpn*—formed two clusters in this dataset. The generation of sequence data could provide a warning to infection control decision-makers and clinical guidance to reduce the burden of CRE through genetic determinants of hypervirulence. While the generation of complete sequences of plasmids can be costly and time-consuming, replicon sequences could be a suitable alternative for the screening of hybrid plasmids.

## Figures and Tables

**Figure 1 antibiotics-13-01224-f001:**
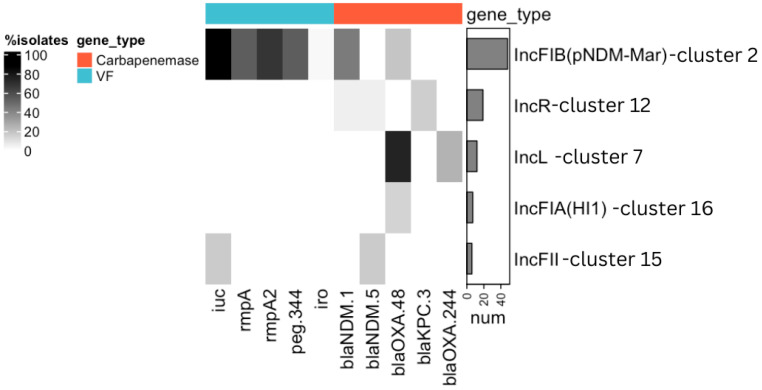
Prevalence of VFs and carbapenemase genes in the identified clusters of plasmids. The clusters of the plasmids were named according to the prevalent replicon gene in this cluster, as shown on the right side of the figure. Only those clusters in which VFs and carbapenemase genes were identified are included in the figure. Genes are grouped by their classification into VFs (blue) or carbapenemase genes (red). Gray shading shows the prevalence of each gene within each cluster. The bar chart on the right side indicates the number of plasmids in each cluster. iuc = *iucABCD* and *iutA* genes; iro = *iroBCDN* genes.

**Figure 2 antibiotics-13-01224-f002:**
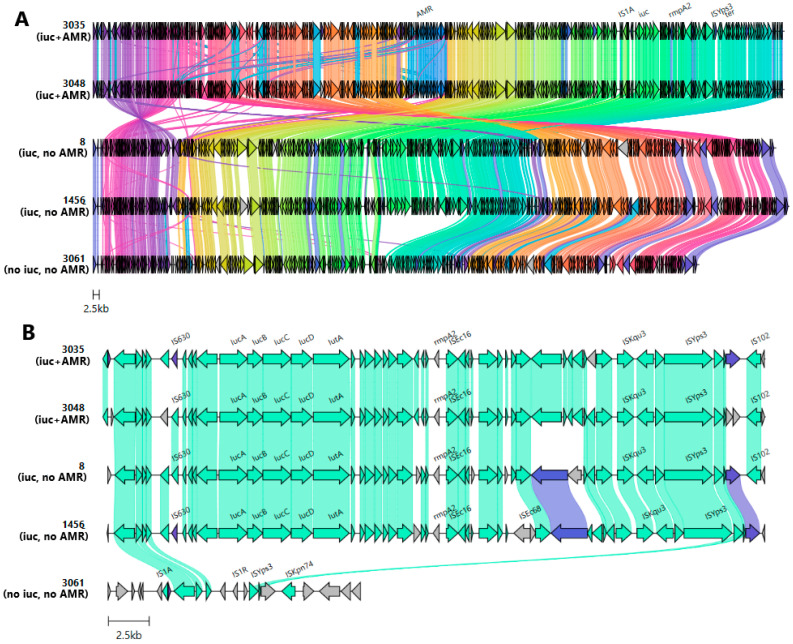
(**A**) Evolution of IncHI1B(pNDM-MAR)/IncFIB(pNDM-Mar)-like plasmids with/without aerobactin. From top to bottom: alignment of plasmids carrying aerobactin and ARGs (pVKpST512_3035, pVKpST512_3048), plasmids carrying aerobactin (pVKpST147_8, pVKpST395_1456), and one plasmid without the aerobactin cluster (pVKpST512_3061). ARGs, *rmpA2*, tellurium resistance genes (*ter*), and the aerobactin gene cluster (*iuc*) with surrounding MGE are marked at the top. (**B**) Comparison of upstream and downstream regions of *rmp* associated with different mobile variants. Arrows represent coding sequences, and those corresponding to genes of interest are labeled. Arrows are colored according to gene clusters, and shading corresponds to regions of similarity (sequence identity ≥ 30%) as identified by clinker. The following were identified: 3035 stands for plasmid pVKpST512_3035, 3048 for pVKpST512_3048, 8 for pVKpST147_8, 1456 for pVKpST395_1456, and 3061 for pVKpST512_3061.

**Figure 3 antibiotics-13-01224-f003:**
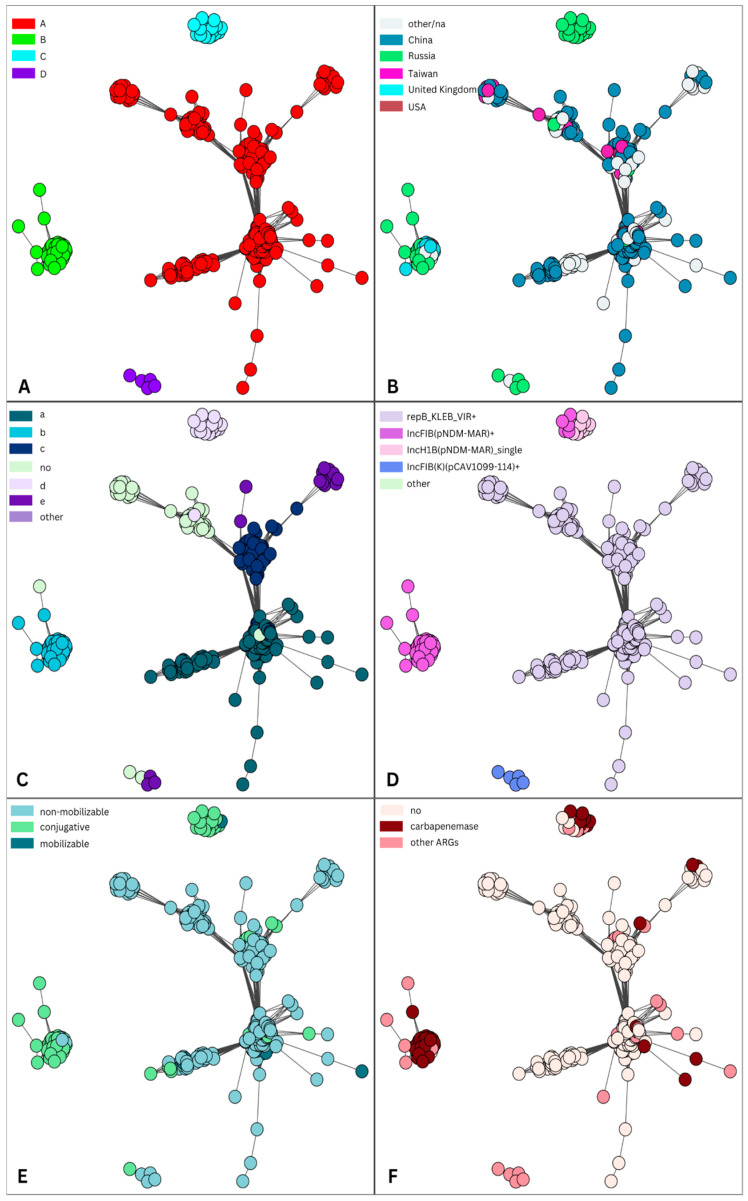
Plasmid similarity network of IncHI1B(pNDM-MAR)/IncFIB(pNDM-Mar)-like plasmids from cluster 2 and the “Global dataset”. For the visual representation of the obtained table with source plasmid node IDs (identification), target plasmid node IDs, and the Jaccard coefficient between them, a graph and a layout for it were generated using the package igraph, where a layout describes the vertical and horizontal placement of nodes when plotting a particular graph structure. Each circle in the figure represents a plasmid, and it is connected with an edge to another node (plasmid) if the Jaccard coefficient between them is greater than 0.8 (see Section 4). Each node (plasmid) is colored according to various features: the assigned cluster (**A**); the country of origin (**B**); the sequence of IncHI1B replicon (**C**); carried replicons, where “+” indicates that a plasmid coded one or more replicons (**D**); mobility (**E**); and ARGs (**F**).

**Figure 4 antibiotics-13-01224-f004:**
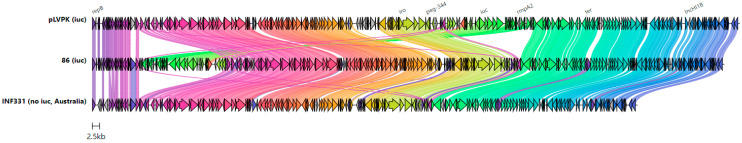
Evolution of plasmids from cluster A with/without aerobactin from our data and the “Global dataset”. From top to bottom: alignment of canonical virulent plasmid pLVPK carrying aerobactin (AY378100), a plasmid carrying aerobactin from our data (86 stands for pVKpST86_86), and plasmid 2 isolate INF331 without the aerobactin cluster from Australia (NZ_LR890425). Replicons, *rmpA*, *rmpA2*, salmochelin (*iro*), and aerobactin gene clusters are marked at the top. Arrows are colored according to gene clusters, and shading corresponds to regions of similarity (sequence identity ≥ 30%) as identified by clinker.

**Table 1 antibiotics-13-01224-t001:** Characteristics of clusters.

Cluster	Plasmids	Species	Replicon	Virulence Factors	Carbapenemase Genes	Number of Plasmids with Genes			Hybrid Plasmids	Predicted Mobility	Median Length, bp
						Carbapenemase	AMR	Virulence			
2	48	*Kpn*, *E. coli*	IncFIB(pNDM-Mar), IncHI1B(pNDM-MAR)	*iutA* (44), *iucA* (44), *iucB* (44), *iucD* (44), *iucC* (43), *rmpA2* (33), *rmpA* (26), *peg-344* (26), *iroBCDN* (1)	*bla*_NDM-1_ (21), *bla*_OXA-48_ (9), *bla*_NDM_ (1)	31	41	44	38	conjugative	321,406
12	19	*Kpn*, *E. coli*	IncR	-	*bla*_NDM-5_ (1), *bla*_NDM-1_ (2), *bla*_KPC-3_ (3)	6	18	0	0	conjugative	51,882
5	13	*Kpn*	ColRNAI	-	-	0	0	0	0	mobilizable	9294
11	13	*Kpn*, *E. coli*	Col(pHAD28)	-	-	0	1	0	0	non-mobilizable	4915
7	12	*Kpn*	IncL	-	*bla*_OXA-48_ (9), *bla*_OXA-244_ (3)	12	12	0	0	conjugative	65,474
13	10	*Kpn*	IncFIB(K)	-	-	0	10	0	0	conjugative	196,937
4	9	*Kpn*	IncFIB	-	-	0	0	0	0	non-mobilizable	109,650
8	8	*Kpn*, *E. coli*	Col(pHAD28), Col440II	-	-	0	0	0	0	mobilizable	3511
16	7	*Kpn*	IncFIA(HI1)	-	*bla*_OXA-48_ (1)	1	1	0	0	conjugative	81,374
3	7	*Kpn*, *E. coli*	Col(MG828)	-	-	0	0	0	0	non-mobilizable	1549
15	6	*Kpn*, *E. coli*	IncFII	*iucABCD*, *iutA* (1)	*bla*_NDM-5_ (1)	1	6	1	1	conjugative	77,238
1	6	*Kpn*	-	-	-	0	0	0	0	non-mobilizable	5010
6	5	*Kpn*	ColpVC	-	-	0	0	0	0	non-mobilizable	1934
17	5	*Kpn*	IncFIB(pQil)	-	-	0	4	0	0	conjugative	141,904
0	5	*Kpn*, *E. coli*	ColRNAI	-	-	0	0	0	0	mobilizable	9716
14	4	*E. coli*	IncFIA, IncI1-I(Alpha)	-	-	0	1	0	0	conjugative	91,854
10	3	*Kpn*, *E. coli*	Col(pHAD28)	-	-	0	1	0	0	mobilizable	4428
9	3	*Kpn*	IncFII	-	-	0	3	0	0	conjugative	81,641

## Data Availability

For all Illumina and MinION reads, assemblies were deposited in the NCBI database under the project accession number PRJNA1133913, and plasmid sequences from the virulence cluster 2 were deposited with GenBank accession PQ126443-PQ126490.

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
