# Peer review of "High-Risk Lineages of Hybrid Plasmids Carrying Virulence and Carbapenemase Genes"

_antibiotics, 2024, doi:10.3390/antibiotics13121224_

Round 1

Reviewer 1 Report

Comments and Suggestions for Authors

The authors have provided relatively new observations which are of importance in virulence space. However, they could really improve data presentation as many figures are not clear in the draft. Additionally, using homology the authors have attempted to define genetic elements responsible for virulence and resistance. Though I agree with this approach, I'd really appreciate if authors could demonstrate experimentally that its these genetic elements which are actively contributing to resistance/virulence.  Their presence itself is primary evidence of possible resistance/virulence while characterizing their activity is different.  I'd tone down the manuscript to reflect this difference if possible.  

Author Response

For research article

Response to Reviewer X Comments

1. Summary

Thank you very much for your time, careful analysis of our work and valuable comments. We made every effort to fix the work. Below we provide answers to your questions.

2. Point-by-point response to Comments and Suggestions for Authors

Comments 1: However, they could really improve data presentation as many figures are not clear in the draft.

Response 1: Thank you for pointing this out. We have revised the figures to improve clarity and ensure they better convey the key findings. Specifically, we have moved Figure 1 to the supplementary materials (now Figure S2) and replaced it with a more straightforward figure that illustrates the prevalence of virulence factors (VFs) and carbapenemase genes in the identified plasmid clusters. Figure 2 is now introduced more clearly in the main text (paragraph 2, page 6, line 173-175). The caption of Figure 3, which shows clusters of plasmids included in the final dataset has been updated with a more detailed description. The final dataset included IncHI1B(pNDM-MAR)/IncFIB(pNDM-MAR)-like plasmids identified in this study and highly similar plasmids retrieved from the NCBI database. Another possible approach to visualizing clustering results is to construct dendrograms. A clustering dendrogram with annotations of analyzed plasmids was added to supplementary files (see Supplementary Figure S4). Additionally, a Microreact project with the obtained dendrogram and metadata was created (https://microreact.org/project/3HndeKYxYKcPGDZKdkgFmu-hybridplasmidsinch1bincfib) and anyone with the link can access the project (paragraph 5.5, page 13, line 468-471).

To better reflect key findings of the study, we decided to correct the title of the paper «High-risk lineages of hybrid plasmids carrying virulence and carbapenemase genes».

Comments 2: Additionally, using homology the authors have attempted to define genetic elements responsible for virulence and resistance. Though I agree with this approach, I'd really appreciate if authors could demonstrate experimentally that its these genetic elements which are actively contributing to resistance/virulence. Their presence itself is primary evidence of possible resistance/virulence while characterizing their activity is different. I'd tone down the manuscript to reflect this difference if possible.

Response 2: Thank you for your valuable remark. We agree with your observation that the presence or absence of virulence genetic determinants is not definitive evidence of phenotypic expression and added this to the discussion (paragraph 3, page 10, line 298-300). Following your response limitations of this study were added to the «Discussion» section (paragraph 3, page 11, line 325-334). However, the association of virulence factors with real virulence potential of the bacteria is well-known. Confirmation of the fact that the genes in each of the dozens of studied strains were indeed expressed and contributed to virulence was beyond the scope of our study work but it could serve as a foundation for a separate comprehensive study. In order to confirm the role of specific virulence determinants in enabling a cell to cause an infection specific to hvKp of hvKp, it is necessary to produce virulence gene knockouts and determine the mutant LD50 in a mouse sepsis model. Conducting such experiments is currently not feasible for our lab. With regard to virulence determinants, the precise set of markers that correlate with phenotypic virulence remains a matter of research to date. However, as this study shows, certain plasmid structures (that might not localize virulence factors) are considered to be a backbone that can be a potential driver (vehicle) for dissemination of genetic determinants high-risk plasmids.

The same applies to antimicrobial resistance genes (ARGs). Although the presence of a resistance gene does not always indicate that a bacterium will exhibit resistance to the corresponding antibiotic, as other factors influencing gene expression may play a role. In most cases, the detection of a specific resistance gene leads to its activity being expressed as an increase in phenotypic resistance to the corresponding class of antibiotics. As the aim of the study was to analyze the diversity, distribution, and genetic features of the plasmids harboring virulence determinants, the analysis of phenotype, despite being technically feasible, is beyond the scope of this research. 

This study showed a large number of identical genetic elements among which are plasmids that do not carry virulence determinants (homologs), but potentially can be carriers based on the mosaic nature of such genetic structures. Since each cluster contained plasmids with/without a virulence cluster, the latter could be considered the precursor of each cluster. To date, most sequencing studies are focused on plasmids carrying specific markers, not considering their relation to plasmids without such genes. Based on the demonstrated evolution of plasmids with/without aerobactin, it is noted that adaptation of plasmids that are inherited along the high-risk clones often occurs before the acquisition of specific determinants, so that analysis of harbored mobile genetic elements is missed. Thus, it is preferable to use an unframed sampling procedure when choosing the isolates for sequencing (paragraph 3, page 10, line 319-321).

Investigation of genetic diversity of plasmids carrying virulence determinants will facilitate our understanding and control of their inter-species spreading as well as spreading of ARGs, which has been frequently associated with the plasmid.

3. Additional clarifications

See the revised manuscript as well as supplementary files attached.

Reviewer 2 Report

Comments and Suggestions for Authors

Dear Authors,

This is wonderful work on plasmidomes and this will help the scientific community to understand the AMR surveillance.

Howver this manuscript needs a minor modifications which I have listed below:

Title: Plasmids carrying virulence determinants of carbapenem resistant K. pneumoniae and E. coli in Russian hospitals 

The title of the article is okay

Introduction: well written and covers the title

Methods and materials:

Line 345 Bacterial Strains section

1.       Kindly provide the distribution of CRE in (blood, urine, broncho-alveolar lavage, bile, sputum, tracheal aspirate, feces, lung, wound, exudate)

2.       How many isolates in total were collected from 19 hopsitals ?

Line 367 Genome assembly and annotation   section

1.       Please reduce the references

Results :  explained in detail

Discussion :

Please write 50 words limitations of this study

Conclusion :  please write 150 words

REFERENCES LINE  NO 449

Reduce the reference under 40.

My conclusion:

Global health is increasingly at risk from carbapenem-resistant K. pneumoniae and E. coli. Because they can cause serious infections and avoid treatment with antibiotics that are used as a last resort, they are important pathogens to keep an eye on and manage. Antibiotic stewardship, strict infection control, and efficient surveillance are crucial in stopping the spread of these resistant microorganisms. Furthermore, to guarantee future CRE infection treatments are effective, funding research for novel therapeutic approaches is essential.

This study will benefit the scientific community at large and can be also helpful in understanding   Horizontal Gene Transfer in K. pneumoniae and E. coli. As plasmidomes are highly dynamic, with plasmids often transferring between strains and species.

Accept for publication after a minor revision.

Comments on the Quality of English Language

na

Author Response

For research article

Response to Reviewer X Comments

1. Summary

Thank you very much for taking the time to review this manuscript. Please find the detailed responses below and the corresponding revisions highlighted in the re-submitted files.

2. Point-by-point response to Comments and Suggestions for Authors

Comments 1: Kindly provide the distribution of CRE in (blood, urine, broncho-alveolar lavage, bile, sputum, tracheal aspirate, feces, lung, wound, exudate).

Response 1: Agree. We have, accordingly, added information about the distribution of CRE in various specimens:

«The collection included isolates from various sources (blood (n=8), broncho-alveolar lavage (n=8), urine (n=6), sputum (n=5), tracheal aspirate (n=5), wound (n=4), lung (n=2), feces (n=1), bile (n=1), exudate (n=1), drainage (n=1))» (paragraph 5.1, page 12, line 412-414).

Comments 2: How many isolates in total were collected from 19 hospitals?

Response 2: Agree. We added information about the total number of isolates:

«In total, 1516 isolates were collected from the 21 hospitals during this period. Among these, 48 Kpn and 5 E. coli clinical isolates were available for whole genome sequencing (WGS)» (paragraph 5.1, page 12, line 408-410).

Comments 3: Line 367 Genome assembly and annotation section. Please reduce the references.

Response 3: We reduced references to 40, including removing some references from Genome Assembly and Annotation sections (paragraph 5.3, page 12, line 424-425).

Comments 4: Discussion: Please write 50 words limitations of this study.

Response 4: We added limitations of the study in the main text:

«Thus, one necessary limitation of our study was sample collection bias since all sequenced isolates were phenotypically filtered by their resistance to carbapenems» (paragraph 3, page 11, line 325-326).

«Another limitation of this study was the inability to validate some characteristics of plasmids (e.g., mobility) experimentally, and prediction of mobility was performed only using computational methods. Additionally, only phenotypic resistance to carbapenems were confirmed experimentally. VFs were identified using the database of the Kleborate tool [24] which consisted of only genetic features with strong evidence of an associated phenotype in Kpn that confirmed clinical relevance based on published experimental data» (paragraph 3, page 11, line 328-334).

Comments 5: Conclusion: please write 150 words.

Response 5: We added a Сonclusion section:

«Our analysis of plasmids in CRE clinical isolates revealed a cluster of plasmids which mainly coded multiple replicons (IncHI1B(pNDM-MAR) and IncFIB(pNDM-Mar)), carbapenemase genes (blaNDM-1  and blaOXA-48) and VFs (iuc, rmpA, rmpA2, peg-344, iro). The comparison analysis of “Global dataset” containing highly identical plasmids from other countries revealed four clusters named in the text by the geographical region from which more than a half of plasmids were collected: "Chinese", "European" and two "Russian" clusters. This might mean that certain backbones of these plasmids had special routes of dissemination. Among them hybrid plasmids, which might be a key element in the formation of a convergent pathotype of Kpn formed two clusters in this dataset. Generation of sequence data could provide a warning to infection control decision makers and clinical guidance to reduce the burden of CRE with genetic determinants of hypervirulence. While generation of complete sequences of plasmids can be costly and time-consuming, replicon sequences could be an available alternative for screening for hybrid plasmids» (paragraph 4, page 12, line 384-398). 

Comments 6: References line no 449. Reduce the reference under 40.

Response 6: Agree. The reference list was reduced as recommended to under 40.

3. Response to Comments on the Quality of English Language

Point 1: The English could be improved to more clearly express the research.

Response 1: Following your response regarding the quality of English, the paper has undergone English language editing by MDPI.

4. Additional clarifications

See the revised manuscript as well as supplementary files attached.

Reviewer 3 Report

Comments and Suggestions for Authors

The manuscript entitlted “Plasmids carrying virulence determinants of carbapenem resistant K. pneumoniae and E. coli in Russian hospitals” by Shapovalova et al., is interesting to read and adds to the scientific knowledge. The manuscript employs a comprehensive use of advanced genomic tools and bioinformatics pipelines, relevant focus on CRE pathogens, a critical global health challenge and strong emphasis on hybrid plasmids carrying both resistance and virulence determinants. All of these are its strength however, the manuscript can benefit from the following;

Major Comments

1.      The dataset is limited by phenotypic filtering for carbapenem resistance and thus potentially excludes the broader diversity in plasmid types and gene profiles. The findings therefore may not generalize to the complete CRE population and as such it is recommended that if possible, the authors should include isolates that are not pre-filtered for carbapenem resistance so as to capture the broader plasmid diversity. If not possible, it is better for the authors to then explicitly discuss the sampling biases and their impact.

2.      The study lacks a thorough comparative analysis between Russian isolates and global datasets and this becomes a weakness with regards to generalization and global significance of the results. It is thus suggested that the authors integrate detailed comparisons with global datasets, particularly emphasizing differences in plasmid diversity, virulence markers, and resistance determinants.

3.      The authors have made conclusions regarding global trends and implications without sufficient supporting data and as such some of the conclusions risk being speculative. It is suggested that the authors reframe conclusions to focus on observed patterns and propose specific future research directions to validate global trends.

4.      The manuscript lacks depth in discussing plasmid evolution and mechanisms of resistance or virulence dissemination and thus readers may struggle to contextualize the findings within a broader biological and epidemiological framework. It is suggested then that the authors expand on the evolutionary role of hybrid plasmids and their implications for resistance spread and virulence enhancement.

5.      The manuscript does not connect findings to practical surveillance or antimicrobial stewardship strategies. This is therefore a missed opportunity to translate findings into actionable public health recommendations. It is proposed that monitoring strategies and policies targeting high-risk plasmid types, particularly hybrid plasmids with virulence and resistance genes be included.

Minor Comments

1.      The abstract is lacking key quantitative details and includes speculative statements and as such must be revised to include metrics such as prevalence rates and percentages while avoiding unsubstantiated claims.

2.      The introduction is repetitive and focuses heavily on general background rather than the novelty of the study. It is suggested that the authors streamline the introduction and highlight the unique contributions of this research.

3.      Several sections present descriptive data without connecting it to broader implications thus providing a more interpretive narrative around major findings, linking them to study objectives and global trends is encouraged.

4.      Computational predictions, such as plasmid mobility, are not validated experimentally so the authors can; 1. Include laboratory validation of key findings or 2. Cite studies supporting the computational methods used.

5.      The conclusion reiterates findings without actionable insights therefore it is suggested that conclusion be strengthened by emphasizing translational impact and proposing future research directions.

6.      Some tables and figures lack sufficient context in the text and may confuse readers therefore the authors must ensure that all visual elements are fully described and directly tied to the discussion.

7.      Some sentences are hanging for example line 119 “section may be divided by subheadings.”

8.      Some sentences such as line 119 “It should provide a concise and precise description of the experimental results, their interpretation, as well as the experimental conclusions that can be drawn.” are misplaced.

Author Response

For research article

Response to Reviewer X Comments

1. Summary

Thank you very much for your time, careful analysis of our work and valuable comments. We made every effort to fix the work. Below we provide answers to your questions.

2. Point-by-point response to Comments and Suggestions for Authors

Major comments:

Comments 1: The dataset is limited by phenotypic filtering for carbapenem resistance and thus potentially excludes the broader diversity in plasmid types and gene profiles. The findings therefore may not generalize to the complete CRE population and as such it is recommended that if possible, the authors should include isolates that are not pre-filtered for carbapenem resistance so as to capture the broader plasmid diversity. If not possible, it is better for the authors to then explicitly discuss the sampling biases and their impact.

Response 1: Limitations of this study were added:

«Thus, one necessary limitation of our study was sample collection bias, as all sequenced isolates were phenotypically filtered by their resistance to carbapenems. Hence, plasmid backbones which could play an important role in the dissemination of VFs in other phenotypes of Enterobacterales isolates were left unexplored. Another limitation of this study was the inability to validate some characteristics of plasmids (e.g., mobility) experimentally, and prediction of mobility was performed only using computational methods. Additionally, only phenotypic resistance to carbapenems were confirmed experimentally. VFs were identified using the database of the Kleborate tool [24] which consisted of only genetic features with strong evidence of an associated phenotype in Kpn that confirmed clinical relevance based on published experimental data» (paragraph 3, page 11, line 325-334).

Comments 2: The study lacks a thorough comparative analysis between Russian isolates and global datasets and this becomes a weakness with regards to generalization and global significance of the results. It is thus suggested that the authors integrate detailed comparisons with global datasets, particularly emphasizing differences in plasmid diversity, virulence markers, and resistance determinants.

Response 2: Comparison between plasmids from this study and highly similar plasmids from other countries were performed from (paragraph 2, page 7-9, line 204-273). A table with included plasmids from the NCBI was added in supplementary files (Supplementary Table S2: Plasmids included in the «Global dataset»). Briefly, clustering based on jaccard coefficients was performed and visualized using networks (in the main text, Figure 3) and a dendrogram (Supplementary Figure S4). In addition, a microreact project was created using this dataset where a table with metadata for all plasmids and a dendrogram were included (paragraph 5.5, page 13, line 468-471).

Comments 3: The authors have made conclusions regarding global trends and implications without sufficient supporting data and as such some of the conclusions risk being speculative. It is suggested that the authors reframe conclusions to focus on observed patterns and propose specific future research directions to validate global trends.

Response 3: We added a conclusion section where results based on analysis of plasmids sequenced in this study and the merged dataset (our data and downloaded sequences from the NCBI) were added:

«Our analysis of plasmids in CRE clinical isolates revealed a cluster of plasmids which mainly coded multiple replicons (IncHI1B(pNDM-MAR) and IncFIB(pNDM-Mar)), carbapenemase genes (blaNDM-1 and blaOXA-48) and VFs (iuc, rmpA, rmpA2, peg-344, iro). The comparison analysis of “Global dataset” containing highly identical plasmids from other countries revealed four clusters named in the text by the geographical region from which more than a half of plasmids were collected: "Chinese", "European" and two "Russian" clusters. This might mean that certain backbones of these plasmids had special routes of dissemination. Among them hybrid plasmids, which might be a key element in the formation of a convergent pathotype of Kpn formed two clusters in this dataset. Generation of sequence data could provide a warning to infection control decision makers and clinical guidance to reduce the burden of CRE with genetic determinants of hypervirulence. While generation of complete sequences of plasmids can be costly and time-consuming, replicon sequences could be an available alternative for screening for hybrid plasmids» (paragraph 4, page 12, line 384-398).

Future directions were added in the Discussion section:

«The present study might serve as a basis for further, more detailed analyses of plasmid epidemiology at the global level. Insights into the diversity of plasmids could be provided through the future widespread application of cost-effective WGS approaches across various countries, combined with the development of novel databases and large-scale analysis tools» (paragraph 3, page 12, line 378-382).

Comments 4: The manuscript lacks depth in discussing plasmid evolution and mechanisms of resistance or virulence dissemination and thus readers may struggle to contextualize the findings within a broader biological and epidemiological framework. It is suggested then that the authors expand on the evolutionary role of hybrid plasmids and their implications for resistance spread and virulence enhancement

Response 4We added a discussion of this topic in the Introduction section:

«The formation of a convergent pathotype—that is, combining MDR and hypervirulence determinants—is the current evolutionary stage of Kpn, which emerged in 2018. Since then, their incidence among various genetic lineages has been increasing globally, which poses a serious threat to the healthcare system. The formation of convergent Kpn is associated with the transition of hypervirulence genes from non-conjugative plasmids into new plasmid backbones and, then, into new Kpn genetic lineages (including European ones). Thus, the spread of resistance genes among Asian genetic lines (CG23) and the main virulence genes among genetic lines of MDR Kpn (ST395, ST147, etс.) was revealed. The mobilization of plasmids with virulence markers led to enhanced horizontal transfer of genes associated with hypervirulence and, as a result, to the formation of new genetic environments. There have been many reports on the acquisition of a single vector carrying a combination of determinants of two Kpn pathotypes, including resistance genes to various classes of antibiotics. Nevertheless, the combination of carbapenem resistance (CR) genes with virulence determinants is of the greatest concern, due to the use of the latter in the treatment of severe infections. The number of large plasmids carrying hypervirulence genes, which often include multiple replicons, has sharply increased. All of them share sequence fragments of the canonical pLVPK plasmid. To date, hybrid plasmids carrying virulence and carbapenemase genes have spread throughout various genetic lines» (paragraph 1, page 2-3, line 96-114).

Comments 5: The manuscript does not connect findings to practical surveillance or antimicrobial stewardship strategies. This is therefore a missed opportunity to translate findings into actionable public health recommendations. It is proposed that monitoring strategies and policies targeting high-risk plasmid types, particularly hybrid plasmids with virulence and resistance genes be included.

Response 5We added a possible strategy to monitor these plasmids:

«It has been suggested [28] that monitoring the core replicon signatures might assist in identifying the spread of these hybrid plasmids. Due to the relative stability of replicon sequences, they could be used as signatures of CRE isolates with virulence potential for clinical management, surveillance, and infection control activities» (paragraph 3, page 12, line 372-376).

Minor comments:

Comments 6:  The abstract is lacking key quantitative details and includes speculative statements and as such must be revised to include metrics such as prevalence rates and percentages while avoiding unsubstantiated claims.

Response 6Quantitative details were added to the abstract:

«The majority of plasmids in the IncHI1B(pNDM-MAR)/IncFIB(pNDM-Mar)-like cluster (79.16%) carried both antibiotic resistance genes (e.g., blaNDM-1, blaOXA-48) and virulence factors (VFs) such as siderophore genes» (abstract, page 1, line 49-52).

We rephrased the last sentences of the abstract to tone down some speculative ideas:

«We hypothesized that hybrid plasmids could play a critical role in the dissemination of antibiotic resistance genes and VFs. Comparative analyses with global plasmid databases revealed high-risk lineages of hybrid plasmids that are predominantly spread throughout Russia at present. Our findings underscore the importance of monitoring plasmid backbones for clinical management, surveillance, and infection control activities» (abstract, page 1-2, line 52-56).

Comments 7:  The introduction is repetitive and focuses heavily on general background rather than the novelty of the study. It is suggested that the authors streamline the introduction and highlight the unique contributions of this research.

Response 7Some details about novelty were streamlined in the Introduction section and have been discussed in the Discussion section where it is stated that up to now, there are a limited number of studies where sequences without genes of interest are also considered and might be an important key element in understanding the process of spreading ARGs and VFs:

«Complete and accurate genome assemblies, which can be produced via hybrid genome assembly of short- and long-read sequencing data, are crucial for the study of plasmid epidemiology. While the majority of studies analyzing human-associated isolates have focused only on resistance determinants and their vehicles, other genetic elements such as VFs located on the plasmids should also be monitored. In this study, we examined the plasmidome of CRE clinical isolates of different STs via genomic analysis, focusing on those sequences which might be related to the transfer of VFs» (paragraph 1, page 3, line 117-124).

«To our knowledge, previous studies of plasmid epidemiology in Enterobacterales have primarily focused on MDR/carbapenemase/aerobactin-carrying sequences [16–18]. We noticed that some plasmids from the virulence cluster which did not carry ARGs and/or VFs were highly similar to those that carried these genes. This finding demonstrates such plasmids can be either precursors of hybrid ones or the result of deletion of ARGs and VFs locuses» (paragraph 3, page 10, line 310-315).

«We hypothesize that VFs could also be considered as acquired high-risk pathogenic factors, which might have the same dissemination mechanisms. This means that, when focusing only on these genes, the opportunity for intervention might be missed; as such, new surveillance frameworks should incorporate previously unselected samples» (paragraph 3, page 10-11, line 317-321).

«Due to the relative stability of replicon sequences, they could be used as signatures of CRE isolates with virulence potential for clinical management, surveillance, and infection control activities» (paragraph 3, page 12, line 374-376).

Comments 8:  Several sections present descriptive data without connecting it to broader implications thus providing a more interpretive narrative around major findings, linking them to study objectives and global trends is encouraged.

Response 8We added a Conclusion section, referred to in Comment 3, where major findings are interpreted in relation to the study objectives and their global relevance (paragraph 4, page 12, line 384-398). Specifically, we discuss how the clustering of plasmids based on replicons, carbapenemase genes, and virulence factors highlights potential dissemination routes and the emergence of hybrid plasmids as key elements in the convergent pathotype of Klebsiella pneumoniae.

Comments 9: Computational predictions, such as plasmid mobility, are not validated experimentally so the authors can; 1. Include laboratory validation of key findings or 2. Cite studies supporting the computational methods used.

Response 9Prediction of mobility was performed only by using a computational approach. We added the tool which was used for this in the section Materials:

«Mob-suite v.3.1.9 [35] was used for plasmid detection and prediction of mobility» (paragraph 5.3, page 13, line 430).

In addition, we tried to reflect uncertainties more accurately, for example by replacing strong “were conjugative” to “predicted to be” in the text (paragraph 2, page 6, line 145; paragraph 3, page 10, line 300).

Comments 10: The conclusion reiterates findings without actionable insights therefore it is suggested that conclusion be strengthened by emphasizing translational impact and proposing future research directions.

Response 10Future research directions were added in the Discussion and in Conclusion sections:

«The present study might serve as a basis for further, more detailed analyses of plasmid epidemiology at the global level. Insights into the diversity of plasmids could be provided through the future widespread application of cost-effective WGS approaches across various countries, combined with the development of novel databases and large-scale analysis tools» (paragraph 3, page 12, line 378-382).

«The generation of sequence data could provide a warning to infection control decision makers and clinical guidance to reduce the burden of CRE with genetic determinants of hypervirulence. While generation of complete sequences of plasmids can be costly and time-consuming, replicon sequences could be an available alternative for screening for hybrid plasmids» (paragraph 4, page 12, line 393-398).

Comments 11: Some tables and figures lack sufficient context in the text and may confuse readers therefore the authors must ensure that all visual elements are fully described and directly tied to the discussion.

Response 11We have revised the figures to improve clarity and ensure they better convey the key findings. Specifically, we have moved Figure 1 to the supplementary materials (now Figure S2) and replaced it with a more straightforward figure that illustrates the prevalence of virulence factors (VFs) and carbapenemase genes in the identified plasmid clusters. Figure 2 is now introduced more clearly in the main text (paragraph 2, page 6, line 173-175). The caption of Figure 3, which shows clusters of plasmids included in the final dataset has been updated with a more detailed description. The final dataset included IncHI1B(pNDM-MAR)/IncFIB(pNDM-MAR)-like plasmids identified in this study and highly similar plasmids retrieved from the NCBI database. Another possible approach to visualizing clustering results is to construct dendrograms. A clustering dendrogram with annotations of analyzed plasmids was added to supplementary files (see Supplementary Figure S4). Additionally, a Microreact project with the obtained dendrogram and metadata was created (https://microreact.org/project/3HndeKYxYKcPGDZKdkgFmu-hybridplasmidsinch1bincfib) and anyone with the link can access the project (paragraph 5.5, page 13, line 468-471).

Comments 12: Some sentences are hanging for example line 119 “section may be divided by subheadings.” Some sentences such as line 119 “It should provide a concise and precise description of the experimental results, their interpretation, as well as the experimental conclusions that can be drawn.” are misplaced.

Response 12: Thank you for pointing it out. These sentences were removed from the text.

3. Additional clarifications

See the revised manuscript as well as supplementary files attached.

Round 2

Reviewer 3 Report

Comments and Suggestions for Authors

I have no further comments